# ADAPTIVE RESOLUTION RESIDUAL NETWORKS

## ABSTRACT

The majority of deep learning methods for signals assume a fixed signal resolution during training and inference, making it impractical to apply a single network at various signal resolutions. We address this shortcoming by introducing *Adaptive Resolution Residual Networks* (ARRNs) that implement two novel components: *Laplacian residuals*, which define the structure of ARRNs and allow compressing high-resolution ARRNs into low-resolution ARRNs, and *Laplacian dropout*, which improves the robustness of compressed ARRNs through a training augmentation. We formulate *Laplacian residuals* by combining the properties of standard residuals and Laplacian pyramids. Thanks to this structure, *lower resolution signals require a lower number of Laplacian residuals for exact computation*. This adaptation greatly reduces the computational cost of inference on lower resolution signals. This adaptation is effectively instantaneous and requires no additional training. We formulate *Laplacian dropout* through the converse idea that *randomly lowering the number of Laplacian residuals is equivalent to randomly lowering signal resolution*. We leverage this as a training augmentation that has the effect of improving the performance of the many low-resolution ARRNs that can be derived from a single high-resolution ARRN. We provide a solid theoretical grounding for the advantageous properties of ARRNs, along with a set of experiments that demonstrate these properties in practice.

## 1 INTRODUCTION

Natural signals such as images, audio, and volumetric scans are acquired by sensor systems whose resolution can vary widely. The majority of deep learning methods for signals disregard the challenges posed by this variety, compromising robustness and computational efficiency for simplicity. These methods are constrained to use identical training resolution and inference resolution, meaning inference at various resolutions requires rediscretizing to the training resolution of the model as a preprocessing step. When inference resolution is lower than training resolution, this wastes the computational advantage that could be gained by leveraging the lower information content of the input and also presents the model with an input that may be out-of-distribution. The methods that tackle this problem directly make the opposite trade-off. These methods support arbitrary inference resolution independently of training resolution but operate on atypical signal representations that are incompatible with most standard layers.

We combine the best features of both approaches by introducing *Adaptive Resolution Residual Networks* (ARRNs), which can adapt to various resolutions easily and robustly thanks to two components: *Laplacian residuals*, which define the structure of ARRNs and allow rediscretization, and *Laplacian dropout*, which improves the robustness of rediscretized ARRNs through a training augmentation.

We formulate *Laplacian residuals* by combining the properties of standard residuals (He et al., 2016b;a) and Laplacian pyramids (Burt & Adelson, 1987). Thanks to this structure, *rediscretizing an ARRN to a lower resolution simply means evaluating a lower number of Laplacian residuals*. This form of rediscretization has many benefits: it improves computational efficiency at lower resolutions; it can be applied instantaneously at inference suit the given resolution; it imposes very few design constraints on the layers nested within Laplacian residuals, unlike prior work based on *neural operators* and *implicit neural representations*, which are incompatible with most layers typically used with signals.

We formulate *Laplacian dropout* through the converse idea that *randomly lowering the number of Laplacian residuals is equivalent to randomly rediscretizing an ARRN to a lower resolution.* We leverage this as a training augmentation that has the effect of improving the robustness of the many low-resolution ARRNs that can be derived from a single high-resolution ARRN.

We situate ARRNs relative to prior works in section 2, provide theoretical analysis for the advantageous properties of ARRNs in section 4, and show a set of experiments that demonstrate these properties in practice in section 5, where we train ARRN and competing methods at a single resolution, then evaluate them at various resolutions.

## 2  RELATED WORKS

In this section, we review related works that allow the formulation of deep learning architectures which can adapt to various resolutions.

**Adaptive resolution through neural operators.**  Li et al. (2020); Kovachki et al. (2021); Fanaskov & Oseledets (2022); Bartolucci et al. (2023) are neural architectures whose inputs and outputs are formalized as *continuous functions* that are encoded and manipulated as *discrete functions*. The layers that form these architectures are conceptualized as *operators on continuous functions* which are translated to *operators on discrete functions* for computation. This allows adaptation to various resolutions in principle but comes at the cost of added complexity in layer design. Each layer must maintain an equivalence between its *continuous operator* form and *discrete operator* form at any resolution for correctness, which is challenging to enforce (Bartolucci et al., 2023). Our method is a type of neural operator, but it escapes the burden of maintaining the equivalence between continuous functions and discrete functions by delegating the task to Laplacian residuals, which allow the layers within to operate at a fixed resolution.

**Adaptive resolution through implicit neural representations.**  Park et al. (2019); Mescheder et al. (2019); Sitzmann et al. (2020); Mildenhall et al. (2021); Chen et al. (2021); Yang et al. (2021a); Lee & Jin (2022); Xu et al. (2022) allow representing the inputs and outputs of neural architectures as *continuous functions* that are encoded implicitly by parameterized *neural functions*. Neural functions are often governed by a learnable parameter space that is shared across different instances of functions, and that decodes individual functions by navigating across a latent space (Park et al., 2019; Mescheder et al., 2019; Chen et al., 2021). Neural functions can be specialized to represent signals in a variety of contexts, and have had great success with volumetric data and images Park et al. (2019); Mescheder et al. (2019); Sitzmann et al. (2020); Mildenhall et al. (2021). These methods lend themselves well to tasks such as adaptive super-resolution (Chen et al., 2021; Lee & Jin, 2022; Yang et al., 2021a), where a neural function is used to approximate a continuous function that is partially observed at an arbitrary input resolution, which can then be evaluated at an unobserved and also arbitrary output resolution. These methods are much more challenging to use in classification tasks, and generally in any task that involves learning maps between distinct continuous functions, as layers need to operate on a latent representation that is far removed from typical signal representations. Even the implementation of convolution is strenuous (Xu et al., 2022).

**Methods similar to Laplacian residuals.**  Singh et al. (2021) incorporates filtering operations within residuals to separate the frequency content of convolutional networks, although it provides no mechanism for adaptive input or output resolution. Lai et al. (2017) uses Laplacian pyramids to solve super-resolution tasks with adaptive *output* resolution, with residuals ordered by *increasing* resolution. This is unlike our method, which is well suited to classification tasks with adaptive *input* resolution, with residuals ordered by *decreasing* resolution.

**Methods similar to Laplacian dropout.**  Huang et al. (2016) implements a form of dropout where the layers nested within residual blocks may be bypassed randomly. This is somewhat similar to Laplacian dropout, however, this is not equivalent to a form of bandwidth augmentation and does not result in improved robustness to various resolutions. This is highlighted in subsection 5.1.

## 3 BACKGROUND

In this section, we provide a short discussion of Laplacian pyramids that helps interpret the formulation of our Laplacian residuals. We assume some familiarity with the theory of signals from the reader (Fourier, 1888; Whittaker, 1915; 1927; Shannon, 1949; Petersen & Middleton, 1962). As in most neural operators methods, we follow the perspective of *continuous signals* rather than *discrete signals*. We conceptualize signals as functions $s : \mathbf{X} \to \mathbb{R}^f$ mapping from a spatial domain $\mathbf{X} \subset \mathbb{R}^d$ to a feature domain $\mathbb{R}^f$.

### 3.1 LAPLACIAN PYRAMIDS

**Definition.** Laplacian pyramids (Burt & Adelson, 1987) are a useful tool for decomposing signals $s$ into $m$ parts according to their frequency content. They are usually formulated through a recurrence relation that relies on Gaussian lowpass filter kernels, but they can also be formulated with Wittaker-Shannon filter kernels $\phi_n^{\text{low}}$ (Whittaker, 1927), which are ideal. This property makes them act as binary masks in the frequency domain. This property also allows us to construct a filter that is fundamentally tied to a specific resolution, in the sense that convolving any signal with it will yield a lower bandwidth signal that can be correctly sampled at that specific resolution without causing errors, in accordance with the Shannon-Nyquist theorem Shannon (1949). In the recursive formulation of Laplacian pyramids, these filters are ordered by decreasing bandwidth, which is synonymous with decreasing resolution. The base case (Equation 1) takes the original signal $s$ as the starting point of recurrence $p_0^{\text{low}}$:

$$p_0^{\text{low}} = s \tag{1}$$

The recursive case takes the preceding lower bandwidth signal $p_{n-1}^{\text{low}}$, forms the next lower bandwidth signal $p_n^{\text{low}}$ (Equation 2), and forms a difference signal $p_n^{\text{diff}}$ (Equation 3) such that both parts sum to the preceding lower bandwidth signal:

$$p_n^{\text{low}} = p_{n-1}^{\text{low}} * \phi_n^{\text{low}} \tag{2}$$

$$p_n^{\text{diff}} = p_{n-1}^{\text{low}} - p_n^{\text{low}} \tag{3}$$

The conditional part (Equation 4) sets what we refer to as the level of the pyramid $p_n$ to the difference signal $p_n^{\text{diff}}$ for all levels except the last one, which is instead set to the lower bandwidth signal $p_n^{\text{low}}$. This ensures all pyramid levels sum to the original signal:

$$p_n = \begin{cases} p_n^{\text{diff}} & \text{if } n \neq m \\ p_m^{\text{low}} & \text{otherwise} \end{cases} \tag{4}$$

The Laplacian pyramid can be seen as a form of signal decomposition that allows us to reconstruct the signal with progressively more bandwidth as we add more difference signals (indexing backwards from the last level of the pyramid):

$$s * \phi_n^{\text{low}} = p_m + p_{m-1} + \cdots + p_{n+1} + p_n \tag{5}$$

**Visualization.** In Figure 1, we summarize this recursive formulation into a simple diagram that shows one step of recursion; this is intended to allow easy comparison with the Laplacian residuals we later illustrate in Figure 3. In Figure 2, we show an example where we decompose a pair of images using a shallow Laplacian pyramid; we later illustrate its Laplacian residual counterpart in Figure 4.

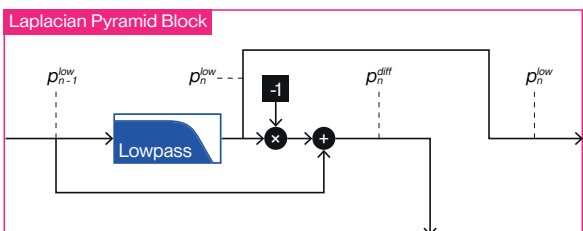

Figure 1: High-level diagram of a single recurrence step of a Laplacian pyramid.

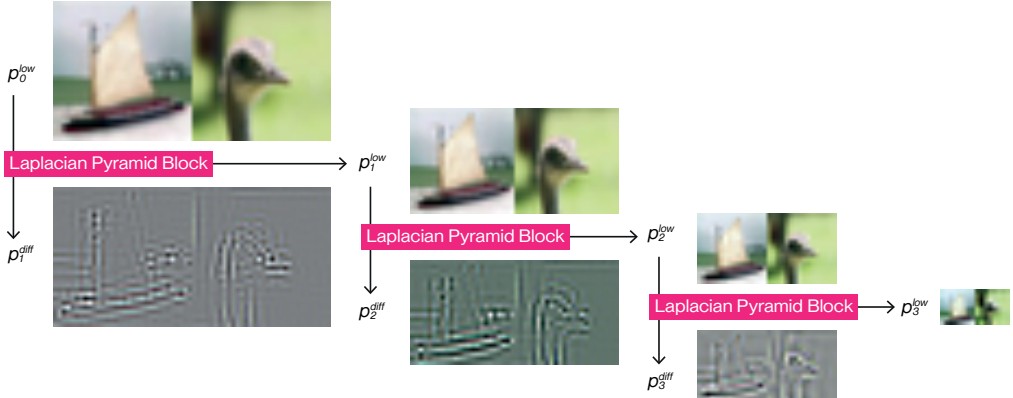

Figure 2: Images decomposed through a Laplacian pyramid. The recursive process starts in the top left with the source image $p_0^{\text{low}}$ and progressively creates lower bandwidth signals $p_{n+1}^{\text{low}}$ moving right, and difference signals $p_{n+1}^{\text{diff}}$ moving down. Together, $p_1^{\text{diff}}$, $p_2^{\text{diff}}$, $p_3^{\text{diff}}$ and $p_3^{\text{low}}$ sum to the original signal $p_0^{\text{low}}$; they are a form of linear decomposition.

## 4 METHOD

In this section, we formulate the two main components of ARRNs: Laplacian residuals (subsection 4.1) and Laplacian dropout (subsection 4.2).

### 4.1 LAPLACIAN RESIDUALS

**Definition.** Laplacian residuals $r_n : (\mathbf{X} \to \mathbb{R}^{f_n}) \to (\mathbf{X} \to \mathbb{R}^{f_{n+1}})$ are composed recursively in a chain of $m$ residuals. Each residual contains some architectural block $b_n : (\mathbf{X} \to \mathbb{R}^{f_n}) \to (\mathbf{X} \to \mathbb{R}^{f_n})$ that can perform any operation as long as its output signal is a constant whenever its input signal is zero (Equation 6, where $a$ is a constant signal); $b_n$ does not need to be conceptualized in the framework of neural operators; $b_n$ does not need the ability to be rediscretized because its discretization is fixed; $b_n$ can be a convolution, transformer, normalization, or composition of multiple layers.

$$b_n(0) = a \tag{6}$$

The filter kernels of Laplacian residuals are set up as in Laplacian pyramids. The base case takes the original signal and performs a linear projection through $\mathbf{A}_0$ to allow a change in feature dimensionality from $f_0$ to $f_1$:

$$r_0 = \mathbf{A}_0 s \tag{7}$$

The recursive case takes the preceding residual $r_{n-1}$, forms a lower bandwidth signal $r_n^{\text{low}}$ (Equation 8), and forms a difference signal $r_n^{\text{diff}}$ (Equation 9):

$$r_n^{\text{low}} = r_{n-1} * \phi_n^{\text{low}} \tag{8}$$

$$r_n^{\text{diff}} = r_{n-1} - r_n^{\text{low}} \tag{9}$$

The difference signal $r_n^{\text{diff}}$ is given to the architectural block $b_n$ contained in the residual. Like in the Laplacian pyramid (Burt & Adelson, 1987), the difference signal $r_n^{\text{diff}}$ explains the gap between two representations of the same signal at different resolutions, one higher, and one lower. We are especially interested in what happens when a signal can be *fully* captured at *either* the higher resolution *or* the lower resolution, meaning a higher resolution representation would be wasteful. We can see that in that case, the difference signal $r_n^{\text{diff}}$ is zero. We want to leverage this by causing a chain of zero terms that we can use for simplifying rediscretization. We do this by using a zero-blocking filter which subtracts the mean:

$$b_n(0) * \phi^{\text{zero}} = 0 \tag{10}$$

We also must perform further filtering with $\phi_n^{\text{low}}$ so that the output conforms to the lower bandwidth signal that the next residual expects as an input. We then apply a skip connection by adding $r_n^{\text{low}}$ to the output, as in standard residuals (He et al., 2016b;a), and apply a linear projection through $\mathbf{A}_n$ to allow a change in feature dimensionality from $f_n$ to $f_{n+1}$ before processing the next residual:

$$r_n = \mathbf{A}_n \left( b_n(r_n^{\text{diff}}) * \phi_n^{\text{low}} * \phi^{\text{zero}} + r_n^{\text{low}} \right) \tag{11}$$

**Rediscretization.** If the spectrum of the signal $s$ is entirely confined within the spectrum of the first filter kernel $\phi_1^{\text{low}}$, then the value of the lower bandwidth residual $r_1^{\text{low}}$ is given by a linear projection of $s$ (Equation 13), and the difference signal $r_1^{\text{diff}}$ is zero (Equation 14). Because the input to the inner architectural block $b_1$ is zero, its output is a constant (Equation 6). Because we then perform filtering with $\phi^{\text{zero}}$, the output of the inner architectural block $b_1$ contributes zero to the residual $r_1$ (Equation 10). The value of the residual $r_1$ is therefore entirely defined by a linear projection of $s$; its exact computation does not need to involve the inner architectural block $b_1$ (Equation 15). We see that this cascade of zeros persists as long as the spectrum of the input signal $s$ is entirely confined within the spectrum of all the lowpass filters $\phi_{n'}^{\text{low}}$ it encounters, with $n' \in [1, n]$:

$$s * \phi_{n'}^{\text{low}} = s \ \forall \ n' \in [1, n] \tag{12}$$

$$\implies r_1^{\text{low}} \qquad = \mathbf{A}_0 s * \phi_1^{\text{low}} = \mathbf{A}_0 s \tag{13}$$

$$\implies r_1^{\text{diff}} \qquad = \mathbf{A}_0 s - \mathbf{A}_0 s = 0 \tag{14}$$

$$\implies r_1 \qquad = \mathbf{A}_1 \left( b_1(0) * \phi_1^{\text{low}} * \phi^{\text{zero}} + \mathbf{A}_0 s \right) = \mathbf{A}_1 \mathbf{A}_0 s \tag{15}$$

$$\vdots$$

$$\implies r_n^{\text{low}} \qquad = \mathbf{A}_{n-1} \cdots \mathbf{A}_0 s * \phi_n^{\text{low}} = \mathbf{A}_{n-1} \cdots \mathbf{A}_0 s \tag{16}$$

$$\implies r_n^{\text{diff}} \qquad = \mathbf{A}_{n-1} \cdots \mathbf{A}_0 s - \mathbf{A}_{n-1} \cdots \mathbf{A}_0 s = 0 \tag{17}$$

$$\implies r_n \qquad = \mathbf{A}_n \left( b_n(0) * \phi_n^{\text{low}} * \phi^{\text{zero}} + \mathbf{A}_{n-1} \cdots \mathbf{A}_0 s \right) = \mathbf{A}_n \cdots \mathbf{A}_0 s \tag{18}$$

We can therefore exactly evaluate $r_n$ by skipping all filters $\phi_{n'}^{\text{low}}$ and all inner architectural blocks $b_{n'}$ with $n' \in [1, n]$, by instead applying a single linear projection with a precomputed matrix $\mathbf{A}_n^{\text{chain}} = \mathbf{A}_n \cdots \mathbf{A}_0$. This allows us to rediscretize high-resolution ARRNs into low-resolution AR-RNs with greater computational efficiency, without performance degradation, and without difficult design constraints.

**Implementation.** We implement all filtering and rediscretization operations using Kaiser-windowed Whittaker-Shannon interpolation (Whittaker, 1915) based on separable polyphase convolutions that effectively extend Smith (2002); Yang et al. (2021b) to higher dimensionality. We apply an approximation that includes 6 zero-crossings, includes no rolloff, uses $\beta = 14.769656459379492$ and uses edge replication padding. We merge consecutive rediscretization operations into single operations based on the analytic interpretation of Whittaker-Shannon filtering.

In Laplacian residuals, the $r_n^{\text{diff}}$ terms of Equation 9 are computed while preserving the original resolution, while the $r_n^{\text{low}}$ and $b_n(r_n) * \phi_n^{\text{low}}$ terms of Equation 11 are computed while rediscretizing to a lower resolution. By following this process, we always use the lowest resolution that allows appropriate representation of the signals. In the experimental setup, all rediscretization needed as a preprocessing step is done through this interpolation method.

**Visualization.** In Figure 3, we illustrate the recursive formulation of Laplacian residuals in a simple block diagram; we can see on the left the same elements that compose the Laplacian pyramid shown in Figure 1. In Figure 2, we show a visualization of a small ARRN by tapping into $r_n^{\text{low}}$, the input to every Laplacian residual, and $r_n^{\text{diff}}$, the input to every architectural block wrapped within a Laplacian residual; this echoes the structure of the Laplacian pyramid shown in Figure 2.

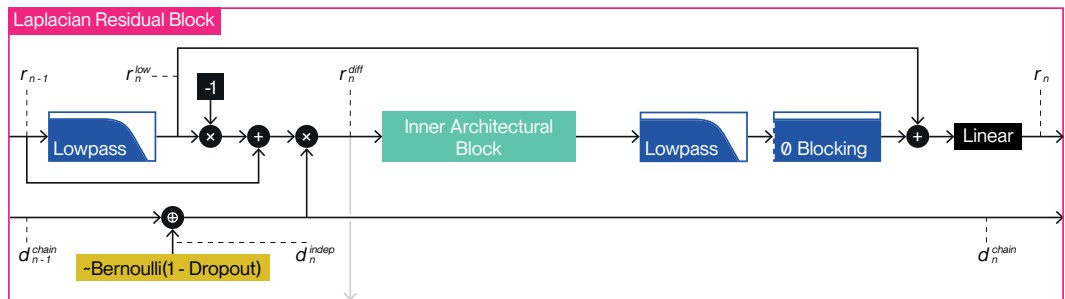

Figure 3: High-level diagram of a Laplacian residual which implements Laplacian dropout.

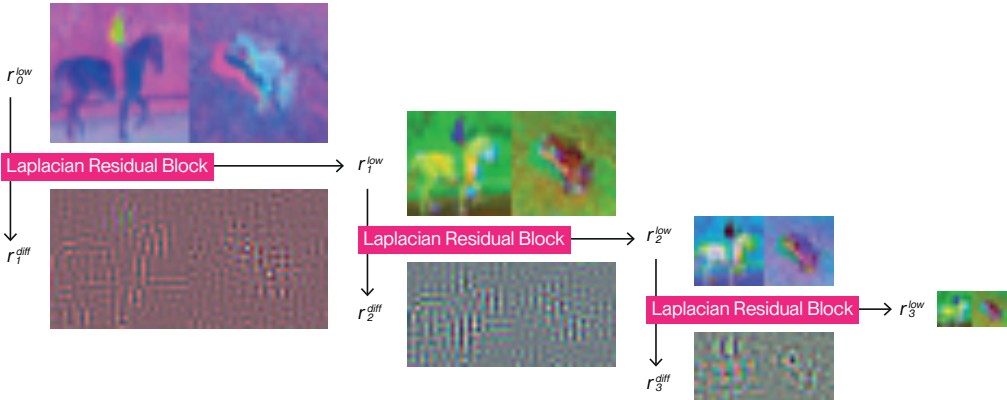

Figure 4: Images of PCA analysis of feature maps created by an ARRN's Laplacian residuals. The process starts in the top left with the source image $r_0$ that has been mapped through $\mathbf{A}_0$. Moving downwards, we see the difference signal $r_{n+1}^{\mathrm{diff}}$ that is later given to the architectural block $b_n$, which is formed in the same way as with Laplacian pyramids. Moving right, we get a lower bandwidth signal $r_{n+1}^{\mathrm{low}}$ based on the output of the last Laplacian residual.

## 4.2 LAPLACIAN DROPOUT

When we show a low-bandwidth signal to a high-resolution ARRN, a set of consecutive early Laplacian residuals may be zero. Conversely, if we show a high-resolution signal to a high-resolution ARRN but randomly zero out a set of consecutive early residuals, this will be equivalent to showing a randomly lowered resolution signal to a correspondingly lowered resolution rediscretized ARRN. Laplacian dropout simply implements this during training to encourage robustness at low resolution. We gate the difference signal $r_n^{\mathrm{diff}}$ with a Bernoulli random variable chained through the logical or operator (Equation 20) to implement the consecutiveness constraint.

$$d_n^{\mathrm{indep}} \sim \mathrm{B}(1 - p_n) \tag{19}$$

$$d_n^{\mathrm{chain}} = d_n^{\mathrm{indep}} \oplus d_{n-1}^{\mathrm{chain}} \tag{20}$$

$$r_n^{\mathrm{diff}} = d_n^{\mathrm{chain}}(r_{n-1} - r_n^{\mathrm{low}}) \tag{21}$$

## 5 EXPERIMENTS

Our experiments demonstrate that rediscretized ARRNs have identical or better performance than non-rediscretized ARRNs; that rediscretized ARRNs have vastly lower inference time than non-rediscretized ARRNs; and that ARRNs are robust to low-resolution signals.

We construct a set of experimental setups each evaluated on the **CIFAR10**, **CIFAR100** (Krizhevsky et al., 2009), **TinyImageNet** (Le & Yang, 2015) and **STL10** (Coates et al., 2011) image classifica-

tion datasets. All models are trained once for 100 epochs at the native dataset resolution ($32 \times 32$ for CIFAR10 and CIFAR100, $64 \times 64$ for TinyImageNet, and $96 \times 96$ for STL10). All models are then evaluated at various lower resolutions. Since the methods we compare do not have the ability to adapt to lower resolutions, the images are rediscretized to the lower resolutions, then rediscretized back to the native dataset resolution during evaluation (see subsection A.1 for an illustrated explanation). Thus, all methods have access to the same information in a fair manner.

We compare our **ARRN** (subsection A.1 explains the architecture design in detail; 5.33M-8.09M for CIFAR10, 9.59M-14.5M for CIFAR100, 15.0M-19.8M for TinyImageNet, and 13.8M-18.4M for STL10 depending on rediscretization) against a wide range of convolutional network families that are well-suited for the classification task we test: **ResNet** (11.1M-42.5M) He et al. (2016b), **WideResNetV2** (66.8M-124M) (Zagoruyko & Komodakis, 2016), **MobileNetV3** (1.52M-4.21M) Howard et al. (2019), and **EfficientNetV2** (20.2M-117.2M).

## 5.1 ROBUSTNESS

We validate the ability of Laplacian dropout to increase the robustness of ARRNs to low-resolution signals. Figure 5 shows that ARRNs with Laplacian dropout (red lines) are vastly superior to ARRNs without Laplacian dropout (black lines), and stronger than all standard methods. We also note that EfficientNetV2 implements a form of residual dropout (Huang et al., 2016) yet does not display improved robustness at low resolution.

## 5.2 REDISCRETIZATION: CORRECTNESS

We confirm that ARRNs can be rediscretized without degrading performance. Figure 5 shows the performance of ARRNs evaluated with rediscretization (full lines), meaning they discard certain Laplacian residuals, and without rediscretization (dashed lines), meaning they always use all Laplacian residuals. For models trained with Laplacian dropout (red lines), the performance of rediscretized ARRNs is identical or better in nearly all cases. For models trained without Laplacian dropout (black lines), the performance of rediscretized ARRNs is worse. This is likely a result of the approximate filters used by the implementation, which allow a small quantity of information to bleed through. This bleed-through is zeroed out both by Laplacian dropout and by rediscretization, which is consistent with the observation that ARRNs trained with Laplacian dropout perform better with rediscretization, and that ARRNs trained without Laplacian dropout perform better without rediscretization.

## 5.3 REDISCRETIZATION: INFERENCE TIME

We confirm the computational savings granted by rediscretization, reusing the previous experimental setup. We perform time measurements by using CUDA event timers and CUDA synchronization barriers around the forward pass of the network to eliminate other sources of overhead, such as data loading, and we sum these time increments over all batches of the full dataset. We repeat this process 10 times and pick the median to reduce the effect of outliers. Figure 6 shows the inference time of ARRNs with rediscretization (full lines) and without rediscretization (dashed lines). As expected, rediscretization reduces inference time at lower resolutions, as a lower number of Laplacian residuals need to be evaluated. Relative to well-engineered standard methods, ARRNs also have reasonable inference time.

## 5.4 REDISCRETIZATION: ADAPTATION TIME

We measure the computational cost that is incurred at the moment we rediscretize a network. This consists in precomputing the matrix product $\mathbf{A}_n^{\text{chain}} = \mathbf{A}_n \cdots \mathbf{A}_0$ outlined in subsection 4.1. We again reuse the previous experimental setup, performing measurements with the same techniques. Figure 7 shows that adaptation takes at most 750 microseconds, which is negligible even for many real-time applications.

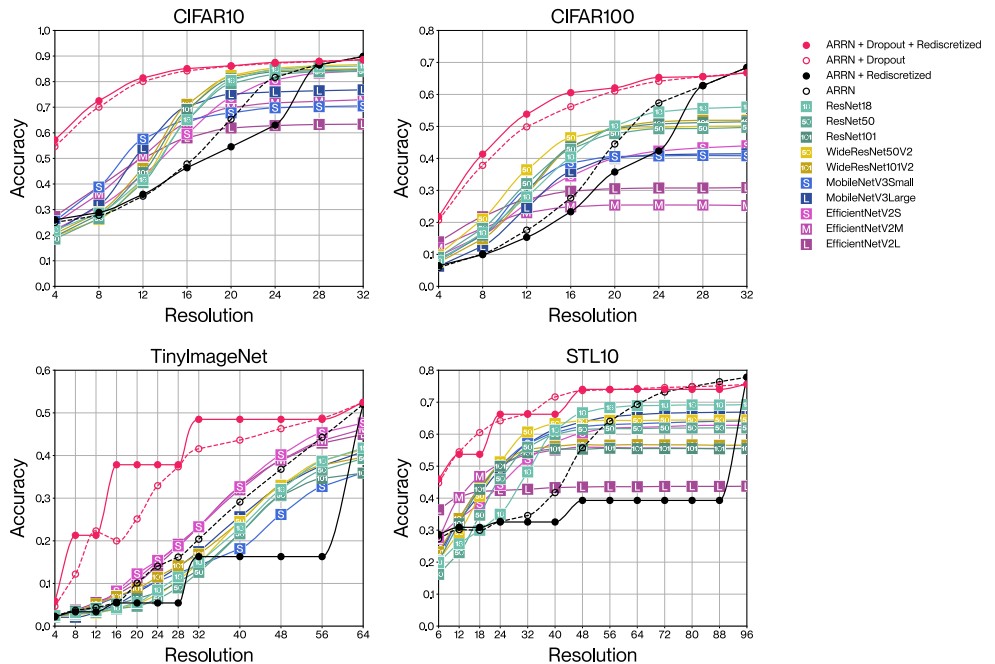

Figure 5: Performance of all methods at changing resolution. Each model is trained at the native dataset resolution and tested at various lower resolutions. Our method ARRN robustly maintains its accuracy at lower resolutions.

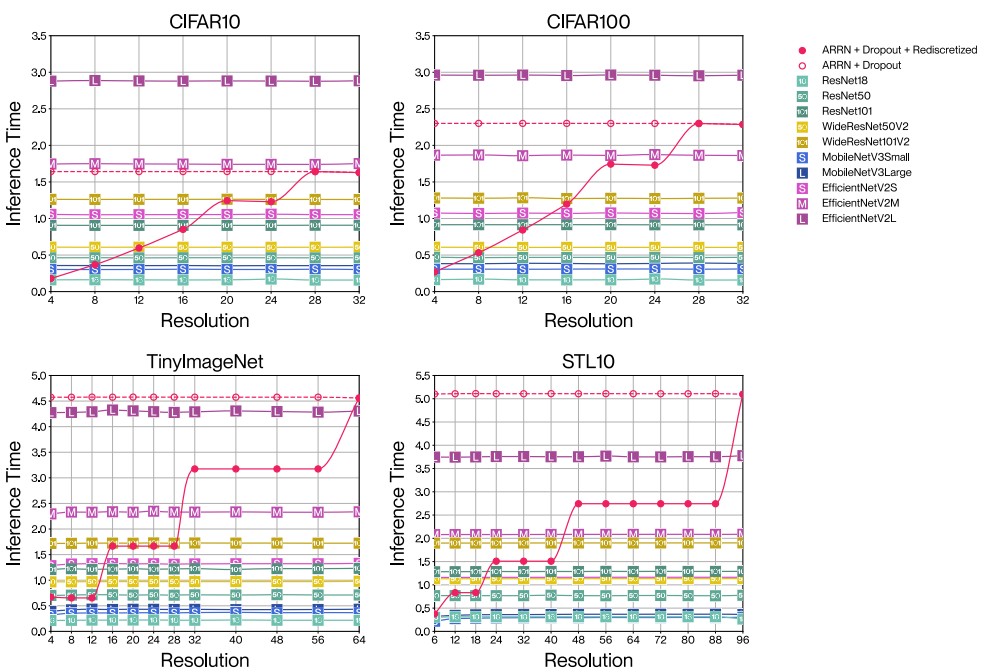

Figure 6: Inference time of all methods at various resolutions. This is summed over the whole dataset for each resolution. Our method ARRN lowers its inference time at lower resolutions thanks to rediscretization, and displays a reasonable inference time relative to typical convolutional neural networks despite not having a highly optimized implementation.

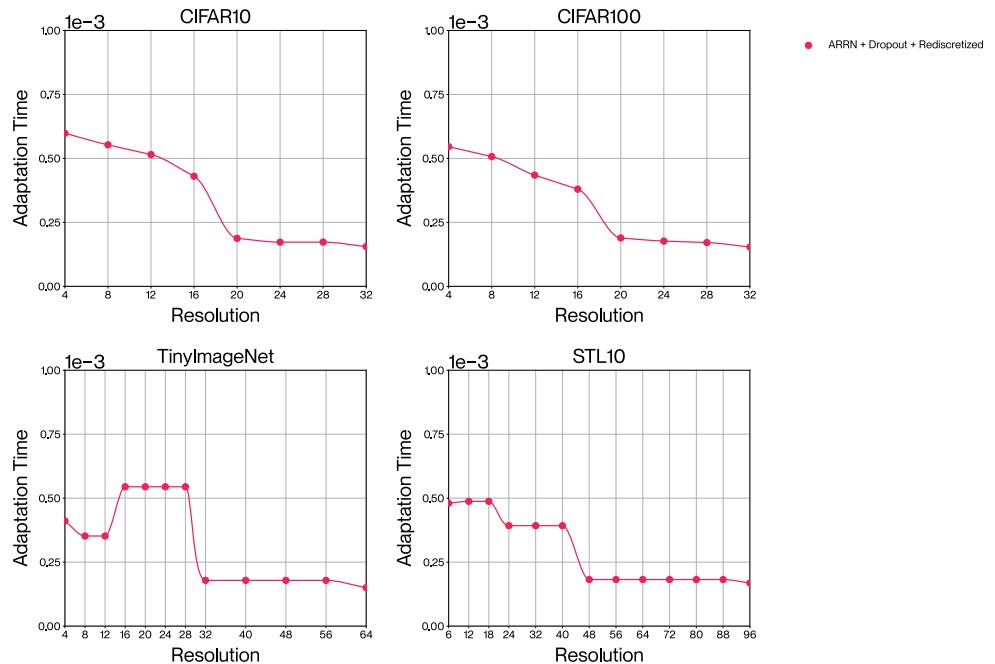

Figure 7: Adaptation time of ARRN at various resolutions. This computation spans less than 750 microseconds and is only required once when adapting to a specific resolution when the first batch is processed.

## 6 DISCUSSION

We have introduced ARRN, a deep learning architecture for tasks involving multidimensional signals which allows constructing adaptive resolution networks that are free from difficult design constraints. ARRNs substitute standard residuals with *Laplacian residuals*, which allows incorporating a wide variety of architectural blocks into networks that can nearly instantly adapt to various resolutions, and that have a drastically lower computational cost at lower resolution. ARRNs also implement *Laplacian dropout*, which greatly promotes robustness to low resolution. These two components allow training high-resolution ARRNs that can then be adapted into robust low-resolution ARRNs on the fly.

**Future Work:** We have provided evidence on image-based tasks, and explored one possible architectural block that can be nested within Laplacian residuals. Many other tasks may be interesting candidates for our method; real-time applications that have a computation budget which varies through time could greatly benefit from the adaptive nature of our method; the extension of our method to adaptive resolution U-Nets for tasks such as segmentation and depth estimation is also possible. Many other architectural blocks are compatible with ARRNs and could be nested within Laplacian residuals.

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
