# A    SUPPLEMENTARY MATERIAL

## A.1    EXPERIMENTS

**Model evaluation.**    In Figure 8, we illustrate how we evaluate networks at various resolutions in our experimental setup, after having trained them at a fixed resolution.

With standard networks, inference must always take place at the training resolution; lower-resolution input signals must be rediscretized to a higher resolution first to be compatible.

With ARRNs evaluated without rediscretization, we follow the process we usually apply with standard networks; the lower-resolution input signal is rediscretized to a higher resolution, and all residuals are used. The grey parts of the illustration are not skipped.

With ARRNs evaluated with rediscretization, lower-resolution input signals go directly to matching lower-resolution residuals, which reduces computational cost. The grey parts of the illustration are skipped. When input signals have a resolution that falls in between the resolution of the residuals, we must first rediscretize the input signals to match either the resolution of the residual above or the resolution of the residual below. For CIFAR10 and CIFAR100, we round up in resolution. For TinyImageNet and STL10, we round down in resolution. We find this offers consistent performance with good inference time.

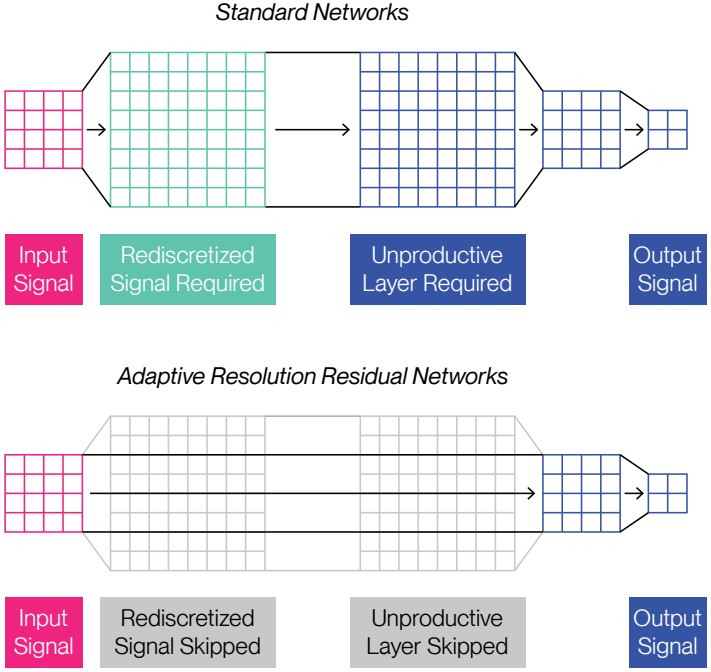

Figure 8: Schematized view showing how standard networks and ARRNs perform inference on lower resolution signals. Each grid shows the resolution of an intermediate signal at some stage in the forward pass of either network; black arrows show the relationship between each intermediate signal in the forward pass; black lines highlight changes in signal resolution. In ARRNs with rediscretization enabled, the intermediate signals faded to grey are skipped.

**Model design.**    The method we propose leaves much freedom for the design of ARRNs; the number of Laplacian residuals, their resolution, their number of features, and their inner architectural block can all be freely picked. The architectural hyperparameters we used in our experiments were found using a series of hand searches and block coordinate searches, maximizing the average accuracy over evaluated resolutions.

We use inner architectural blocks that take inspiration from the parameter-efficient convolutional layers that are used within MobileNetV2 (Sandler et al., 2018) and EfficientNetV2 (Tan & Le, 2021), illustrated in Figure 9, Figure 10, Figure 11, Figure 12. We use a string of 2, 3, 4 and 4 depthwise $3 \times 3$ convolutions for CIFAR10, CIFAR100, TinyImageNet and STL10 respectively, each separated with pointwise $(1 \times 1)$ convolutions. All depthwise convolutions use edge replication padding in order to satisfy Equation 6 and ensure resolution remains fixed within Laplacian residuals. The whole string of convolutions is preceded by a pointwise convolution that expands the feature channel count by a factor of 8, 8, 4 and 8 for CIFAR10, CIFAR100, TinyImageNet and STL10. This is terminated by a pointwise convolution that contracts the feature channel count inversely by the same factor to restore the original feature channel count. Each convolution is separated by a batch normalization (Ioffe & Szegedy, 2015) and a SiLU activation function (Elfwing et al., 2018), chosen for its tendency to produce fewer aliasing artifacts.

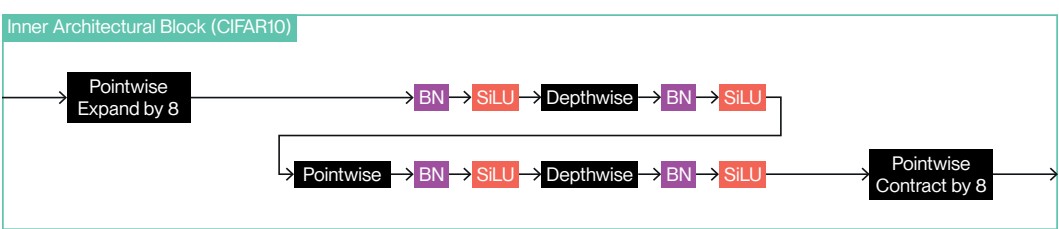

Figure 9: High-level diagram of an inner architectural block nested within a Laplacian residual, in the CIFAR10 ARRN.

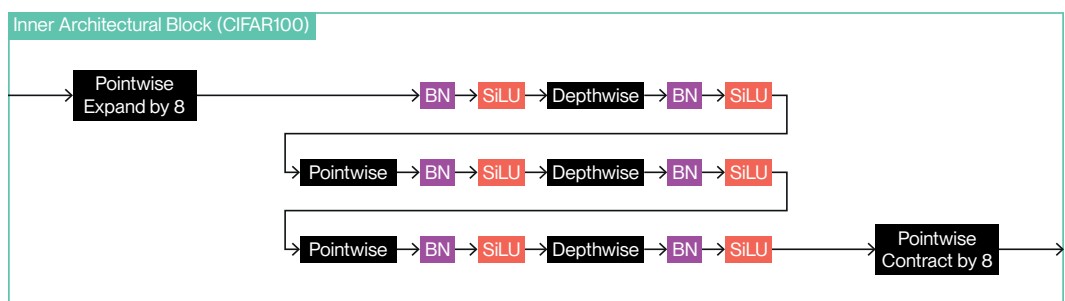

Figure 10: High-level diagram of an inner architectural block nested within a Laplacian residual, in the CIFAR100 ARRN.

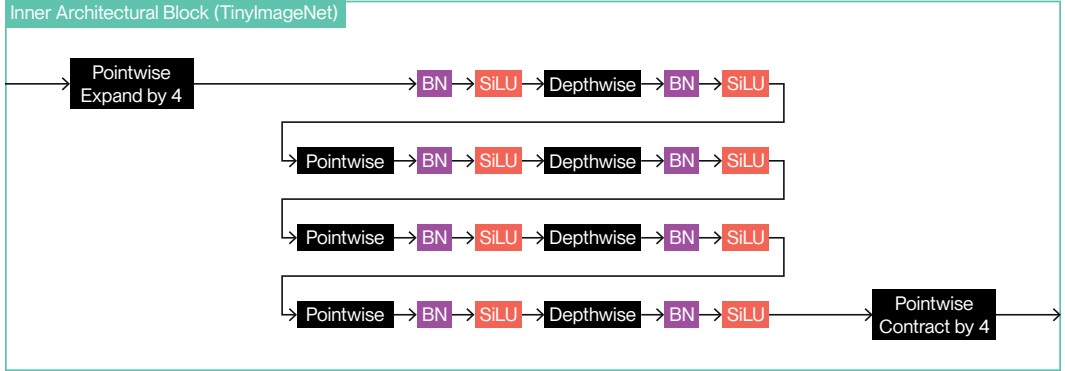

Figure 11: High-level diagram of an inner architectural block nested within a Laplacian residual, in the TinyImageNet ARRN.

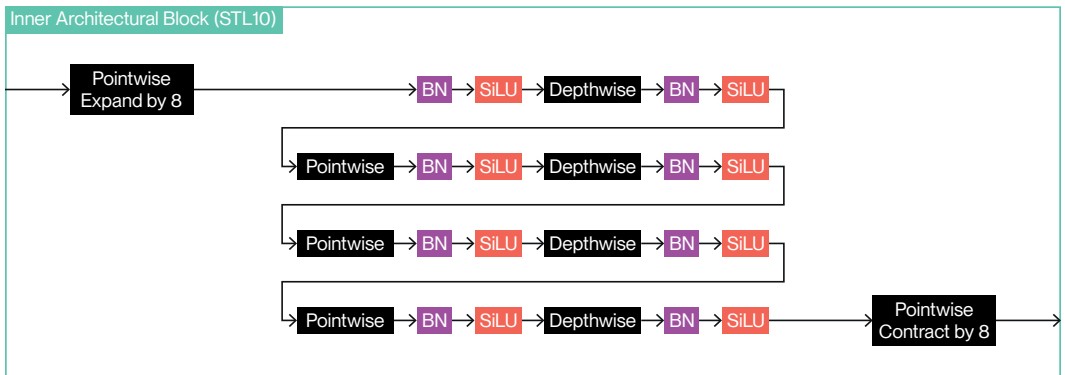

Figure 12: High-level diagram of an inner architectural block nested within a Laplacian residual, in the STL10 ARRN.

For CIFAR10 and CIFAR100, we settled with 6 Laplacian residual blocks of resolution $32 \times 32, 24 \times 24, 16 \times 16, 12 \times 12, 8 \times 8, 4 \times 4$ with feature channel counts of $32, 48, 64, 96, 128, 256$. When enabled, we use a Laplacian dropout rate of $0.6$ and $0.3$ on CIFAR10 and CIFAR100 respectively.

For TinyImageNet, we pick 4 Laplacian residual blocks of resolution $64 \times 64, 32 \times 32, 16 \times 16, 8 \times 8$ with feature channel counts of $32, 128, 256, 512$. When enabled, we use a Laplacian dropout rate of $0.3$.

For STL10, we use 5 Laplacian residual blocks of resolution $96 \times 96, 48 \times 48, 24 \times 24, 12 \times 12, 6 \times 6$ with feature channel counts of $16, 32, 64, 128, 256$. When enabled, we use a Laplacian dropout rate of $0.3$.

**Model training.** For CIFAR10 and CIFAR100, across all methods, we use AdamW (Loshchilov & Hutter, 2017) with a learning rate of $10^{-3}$ and $(\beta_1, \beta_2) = (0.9, 0.999)$, cosine annealing (Loshchilov & Hutter, 2016) to a minimum learning rate of $10^{-5}$ in 100 epochs, weight decay of $10^{-3}$, and a batch size of $128$. We use a basic data augmentation consisting of normalization, random horizontal flipping with $p = 0.5$, and randomized cropping that applies zero-padding by 4 along each edge to raise the resolution, then crops back to the original resolution.

For TinyImageNet and STL10, across all methods, we use SGD with a learning rate of $10^{-2}$, cosine annealing (Loshchilov & Hutter, 2016) to a minimum learning rate of $0$ in 100 epochs, weight decay of $10^{-3}$, and a batch size of $128$. We use TrivialAugmentWide (Müller & Hutter, 2021) to augment training.