# OpenReview forum: "Adaptive Resolution Residual Networks"
_ICLR.cc/2024/Conference — Submitted to ICLR 2024_

### Official Review · Reviewer_Ag6R · 2023-10-28

**Soundness:** 3 good
**Presentation:** 3 good
**Contribution:** 2 fair
**Rating:** 3
**Confidence:** 4

**Summary:**

The paper proposes the ARRN, which is a network architecture designed to address the problem of signal resolution variation in machine learning tasks. It uses Laplacian residual connections to adapt the resolution of models quickly, reducing computation as resolution decreases. ARRNs also incorporate Laplacian dropout to enhance the networks' robustness to low-resolution signals. This allows for training high-resolution ARRNs that can later be compressed into effective low-resolution ARRNs.

**Strengths:**

1.	The idea is simple and clear.
2.	The experiments show the effectiveness of the proposed algorithm.
3.	The proposed algorithm has low computational cost and without requiring retraining.

**Weaknesses:**

1.	The practicability of the algorithm is not very reliable. One issue is that the algorithm can only deal with the sizes contained in the feature maps. The other is that the performance of the algorithm will drop quickly without residual dropout.
2.	The experiments cannot show the advantages of the proposed approach. See the questions below.

**Questions:**

1.	The application scenario of the algorithm is not clear. Because the proposed algorithm can only deal with the resolutions contained in feature maps, and the paper only verifies it on very small images (e.g, 32x32), it cannot explain the effectiveness of the algorithm.
2.	Without the residual dropout, the effectiveness of the algorithm will be poor. This is related to the content described in subsection 4.2. It is not clear whether residual dropout leads to better performance or the laplacian residuls introduced in subsection 4.2.
3.	The paper does not compare with networks that adapt to resolution, but only with the approach of upsampling the low-resolution inputs. This cannot explain the superiority of the algorithm.
4.	How to set the filter kernels \phi^{low}_n?
5.	There is a missing of reference in page 1. In the first paragraph of subsection 3.1, there are two successive “ideal”.

---

> ### Author Response · Authors · 2023-11-17
>
> We appreciate your engagement with our work. We have provided a revised manuscript that includes more thourough experiments, that is more concise, and that better situates and differentiates our method relative to prior methods.
>
> **Q1.1)** The application scenario of the algorithm is not clear. The proposed algorithm can only deal with the resolutions corresponding to those of Laplacian residuals.
>
> **A1.1)** As you have noted, rediscretization applies in resolution steps that correspond to the resolutions of Laplacian residuals. While not ideal, we believe this is in fact an excellent compromise. Given an input whose resolution falls just between the resolution of two Laplacian residuals, the input can simply be rediscretized to the resolution of either residual block in a preprocessing step, which still results in excellent performance and improved computational efficiency, as shown in our experiments. This compromise is what allows us to make the resolution of layers nested within Laplacian residual blocks fixed while making the resolution of the whole network adaptive. This allows us to use standard layers, unlike previous methods that feature adaptive input resolution. This is a central aspect of the novelty of the method, and it makes it much easier to integrate with existing work on convolutional neural networks.
>
> **Q3)** The paper does not compare with networks have adaptive input resolution.
>
> **A3)** We do not see this as a detraction to the novelty of our method. ARRN is a drop-in technique that provides adaptive input resolution for residual networks built with standard layers. It makes most sense to compare ARRN with typical convolutional neural networks that are constructed with similar standard layers, as it reveals that our drop-in technique causes no loss in performance. This comparison cannot be made against existing methods with adaptive input resolution methods because they are not drop-in techniques. This is because they impose difficult design constraints on the layers they contain, which is explained in detail in the updated related works section. In contrast, our method is free from these limitations.
>
> **Q2)** The method performs poorly without Laplacian dropout. It is not clear whether Laplacian dropout is responsible for improved robustness at low resolution.
>
> **A2)** The effectiveness of Laplacian dropout is supported by our experiments.  With Laplacian dropout, rediscretized ARRNs show overwhelming improvement in robustness at low resolution without compromise in performance at high resolution, even compared to well-established convolutional neural networks that specialise at a single high resolution. Without Laplacian dropout, rediscretized ARRNs show very poor robustness at low resolution. This poor performance is not an indication that the method is flawed, but an indication that Laplacian dropout is responsible for the improvement in robustness. This poor performance is without practical consequence because Laplacian dropout will be employed in every use case that aligns with the core motivation of the method, which is to allow adapting high resolution ARRNs to low resolution ARRNs for improved computational efficiency without re-training.
>
> **Q4)** How are the filter kernels set?
>
> **A4)** We have updated the implementation paragraph of subsection 4.1 to provide detailed parameters for the kernels.
>
> **Q1.2)** The experiments only cover small resolution datasets.
>
> **A1.2)** We have updated our experiments to cover the higher resolution TinyImageNet dataset, and also include a larger set of baseline methods spanning all usual variants of ResNet, WideResNetV2, MobileNetV3, and EfficientNetV2. Our method ARRN remains more robust at lower resolution than these standard methods while also beating their performance at the highest resolution. We would like to include the ImageNet dataset in our experiments however given limited time and computational resources we are not able to cover it appropriately. We believe our contribution is solid nonetheless as the novel properties of ARRNs are well supported by theory.
>
> We hope that this comment, the improved manuscript, and the updated set of experiments address your concerns.

---

> > ### Author Response · Authors · 2023-11-20
> >
> > Dear Reviewer,
> >
> > We hope that you've had a chance to read our responses and clarification. As the end of the discussion period is approaching, we would greatly appreciate it if you could confirm that our updates have addressed your concerns.

---

> > > ### Author Response · Authors · 2023-11-23
> > >
> > > Dear reviewer,
> > >
> > > We have just provided a second revision of our experiments that includes the higher-resolution STL-10 dataset. Our method ARRN retains a significant advantage in accuracy at all resolutions against all baselines.
> > >
> > > As the discussion period is coming to an end, we would appreciate that you clarify whether the revised manuscript, improved experiments, and answers to your questions have addressed your concerns.

---

### Official Review · Reviewer_DCC2 · 2023-10-31

**Soundness:** 2 fair
**Presentation:** 2 fair
**Contribution:** 2 fair
**Rating:** 3
**Confidence:** 5

**Summary:**

To address most deep learning methods for signals that assume a fixed signal resolution during training and inference, this paper proposes Adaptive Resolution Residual Networks (ARRNs). The networks include Laplacian residuals and Laplacian dropout. The Laplacian residuals allow the compressing of high-resolution ARRNs into low-resolution ARRNs, and the Laplacian dropout improves the robustness of compressed ARRNs through training augmentation. The experiments demonstrate the effectiveness of the proposed method.

**Strengths:**

This paper proposes Adaptive Resolution Residual Networks (ARRNs) which can be used at various signal resolutions. To this end, the authors propose Laplacian residual and Laplacian dropout.

**Weaknesses:**

1. The authors do not investigate sufficient related work.
2. The experiments section should be improved.

**Questions:**

1. Some related studies are necessary.  The authors state that "The majority of deep learning methods for signals assume a fixed signal resolution during training and inference" There are many networks [1-4] that do not have such an assumption and they can be trained with different resolution inputs. It would be better to discuss these papers.

[1] Learning continuous image representation with local implicit image function.
[2]  Local texture estimator for implicit representation function.
[3] Implicit transformer network for screen content image continuous super-resolution.
[4] CiaoSR: Continuous Implicit Attention-in-Attention Network for Arbitrary-Scale Image Super-Resolution.

2. The authors highlight that it is impractical for the majority of deep learning methods to apply a single network at various signal resolutions. Could you provide some examples of such application scenarios in the paper?

3. The experiments are not sufficient. The authors mainly conduct experiments on image classification on CIFAR10 and CIFAR100. It is not convincing that the proposed method does not hinder performance or require re-training. The authors should conduct more experiments on ImageNet with more neural networks. In addition, can the ARRNs used in other tasks, e.g., super-resolution?

4. On the first page, in the third paragraph of the introduction, there is an issue: illustrated in ??

---

> ### Author Response · Authors · 2023-11-17
>
> We appreciate your engagement with our work. We have provided a revised manuscript that includes more thourough experiments, that is more concise, and that better situates and differentiates our method relative to prior methods.
>
> **Q3.1)** The experiments are not sufficient.
>
> **A3.1)** The updated experiments cover the higher resolution TinyImageNet dataset, and also include a larger set of baseline methods spanning all usual variants of ResNet, WideResNetV2, MobileNetV3, and EfficientNetV2. Our method ARRN remains more robust at lower resolution than these standard methods while also beating their performance at the highest resolution. We would like to include the ImageNet dataset in our experiments however given limited time and computational resources we are not able to cover it appropriately. We believe our contribution is solid nonetheless as the novel properties of ARRNs are well supported by theory.
>
> **Q3.2)** Can ARRNs be used for other tasks such as super-resolution?
>
> **A3.2)** We note that Lai 2017 proposes a form of inverse Laplacian residual block which has been highly successful at adaptive super-resolution tasks. [Edited to cite correct reference] Whereas our Laplacian residuals are ordered by *decreasing* resolution, and can adapt their *input* resolution by removing residuals at the *head* of the chain, inverse Laplacian residuals are ordered by *increasing* resolution, and can adapt their *output* resolution by omitting residuals at the *tail* of the chain. This inverse Laplacian residual block can in principle be combined with our Laplacian residual block to form U-Nets that map arbitrary resolution signals to arbitrary resolution signals, which may be useful in tasks such as segmentation and depth estimation.
>
> **Q1)** Some related studies are necessary.
>
> **A1)** We thank you for pointing our attention towards an important class of methods we had  not considered in our related works section. We have added a discussion of implicit neural representations which overviews the concepts that are central to their formulation, and which highlights the way they differ from neural operator-based methods and our own method.
>
> **Q2)** The authors highlight that it is impractical for the majority of deep learning methods to apply a single network at various signal resolutions. Could you provide some examples of such application scenarios in the paper?
>
> **A2)** We have updated our introduction to better acknowledge methods that allow arbitrary resolution, and to give a concrete example for the impracticality of fixed resolution methods used in a variable resolution context. We are most interested in the scenario where a network trained at high resolution performs inference at low resolution.
>
> We hope that this comment, the improved manuscript, and the updated set of experiments address your concerns.

---

> > ### Author Response · Authors · 2023-11-20
> >
> > Dear Reviewer,
> >
> > We hope that you've had a chance to read our responses and clarification. As the end of the discussion period is approaching, we would greatly appreciate it if you could confirm that our updates have addressed your concerns.

---

> > > ### Author Response · Authors · 2023-11-23
> > >
> > > Dear reviewer,
> > >
> > > We have just provided a second revision of our experiments that includes the higher-resolution STL-10 dataset. Our method ARRN retains a significant advantage in accuracy at all resolutions against all baselines.
> > >
> > > As the discussion period is coming to an end, we would appreciate that you clarify whether the revised manuscript, improved experiments, and answers to your questions have addressed your concerns.

---

### Official Review · Reviewer_FCGS · 2023-10-31

**Soundness:** 2 fair
**Presentation:** 3 good
**Contribution:** 1 poor
**Rating:** 3
**Confidence:** 4

**Summary:**

This paper presents a resolution adaptive network that compress high-resolution image to low-resolution, they also propose Laplacian residuals. The authors claim that the proposed structure, namely residual plus Laplacian pyramids, greatly reduce computational cost on low resolution signals.

**Strengths:**

Using laplacian residual to replace standard residuals in cnn is an interesting and reasonable approach.
The writing is ok.

**Weaknesses:**

1. The most fatal weakness is the experiments. As a paper that specifically discusses image resolution of neural networks, the largest resolution in the experiments is 32 \times 32. The cifar is not a suitable dataset to evaluate ARRN
Notice that a standard ImageNet setting has resolution of 224 \times 224, and many sota works have trained and evaluated on the resolution of 384 \times 384.

2. The novelty is limited. As said in the related work, this work is merely an extension of Lai 2017 & Singh 2021, which applies Laplacian pyramids to signal resolution.

3. The inference time in Figure 7, from my experience, is overwhelming, given the resolution is 32 \times 32 at most. I'm not sure if this is the implementation or hardware problem, therefore i suggest the author provide a comparison between standard residual and laplacian residual.

**Questions:**

See weakness.
Page 1, second last row has an incorrect citation.

---

> ### Author Response · Authors · 2023-11-17
>
> We appreciate your engagement with our work. We have provided a revised manuscript that includes more thourough experiments, that is more concise, and that better situates and differentiates our method relative to prior methods.
>
> **Q2)** The novelty is limited. As said in the related works section, this work is merely an extension of Lai 2017 & Singh 2021.
>
> **A2)** We maintain that Singh 2021 is unrelated to adaptive resolution approaches, and that ARRN is fundamentally distinct from Lai 2017. [Edited to discuss references separately] This prior method uses residuals chained by *increasing* resolution, and can adapt its *output* resolution by omitting residuals at the *tail* of the chain, which is well suited for super-resolution tasks. In contrast, ARRN uses residuals chained by *decreasing* resolution, and can adapt its *input* resolution by removing residuals at the *head* of the chain, which is more appropriate for classification tasks. Removing residuals at the *head* of the chain is nontrivially different from removing residuals at the *tail* of the chain, as later residuals must not be disturbed by the removal. We construct ARRNs around a novel theoretical analysis of this constraint.
>
> We also highlight that prior methods which allow adaptive *input* resolution are either based on *neural operators* or *implicit neural representations*, which operate on unusual signal representations that are incompatible with most standard deep network layers. This covered in detail in the revised related works section of the paper. In contrast, ARRN uses the most ubiquitous form of signal representation, and is compatible with almost all standard deep learning layers. We believe this is key to the usefulness of our method to the community, as it allows easily creating adaptive resolution methods that can leverage a wide range of existing techniques.
>
> **Q1)** The largest resolution in the experiments is 32 \times 32. The CIFAR10 dataset is not suitable to evaluate ARRN.
>
> **A1)** We agree that evaluating methods across a larger space of resolutions is important. We have updated the experiments to cover the higher resolution TinyImageNet dataset, and also include a larger set of baseline methods spanning all usual variants of ResNet, WideResNetV2, MobileNetV3, and EfficientNetV2. Our method ARRN remains more robust at lower resolution than these standard methods while also beating their performance at the highest resolution. We would like to include the ImageNet dataset in our experiments however given limited time and computational resources we are not able to cover it appropriately. We believe our contribution is solid nonetheless as the novel properties of ARRNs are well supported by theory.
>
> **Q3)** The inference time seems excessive. How does ARRN compare to prior works in this respect?
>
> **A3)** The inference time of all methods is now included in Figure 6, as you requested. We apologize if it was not clear that we display the inference time not for *a single batch*, but for the *sum of all batches across the whole dataset*, excluding data loader overhead. While our implementation could be improved upon, ARRN achieves a reasonable inference time relative to prior methods that have had years of tuning from the community, and that are not burdened by the constraints imposed by adaptive resolution.
>
> Let us know if these updates to not address your concerns.

---

> > ### Author Response · Authors · 2023-11-20
> >
> > Dear Reviewer,
> >
> > We hope that you've had a chance to read our responses and clarification. As the end of the discussion period is approaching, we would greatly appreciate it if you could confirm that our updates have addressed your concerns.

---

> > > ### Author Response · Authors · 2023-11-23
> > >
> > > Dear reviewer,
> > >
> > > We have just provided a second revision of our experiments that includes the higher-resolution STL-10 dataset. Our method ARRN retains a significant advantage in accuracy at all resolutions against all baselines.
> > >
> > > As the discussion period is coming to an end, we would appreciate that you clarify whether the revised manuscript, improved experiments, and answers to your questions have addressed your concerns.

---

### Official Review · Reviewer_quBW · 2023-11-01

**Soundness:** 4 excellent
**Presentation:** 4 excellent
**Contribution:** 3 good
**Rating:** 6
**Confidence:** 2

**Summary:**

This paper proposes Adaptive Resolution Residual Networks (ARRNs) to process signals at different resolutions. The proposed method contains two components: Laplacian residuals and Laplacian dropout. The authors show effectivenss of the proposed method mainly on CIFAR dataset at relatively low resolution.

**Strengths:**

1. Having a single network to process images at various resolutions is an important research topic.
2. The paper is well-written and figures are clear. The proposed method is well-motivated, lower resolution signals require a lower number of Laplacian residuals provides a natural and intuitive way to reduce the computational cost.
3. The proposed method is compared to many different prior architectures.

**Weaknesses:**

1. The experiments only show results for images at resolution less than 32, which is not very practical as real-world images to process are usually at much higher resolutions in applications.
2. The current experiments are mainly on CIFAR, more diverse datasets could make the results more convincing.
3. Typo: page 1, bottom - "illustrated in ??"

**Questions:**

What are the main challenges to make the proposed method work for images at a more practical resolution? (e.g. around 256 in ImageNet)

---

> ### Author Response · Authors · 2023-11-17
>
> We thank you for your engagement with our work and for the feedback you have provided.
>
> **Q1/Q2)** The experiments feature datasets that have relatively small resolution and that could be more diverse. Improving this would make the results more convincing.
>
> **A1/A2)** We have revised our work to include an improved set of experiments that cover the higher resolution TinyImageNet dataset, and that also include a larger set of baseline methods spanning all usual variants of ResNet, WideResNetV2, MobileNetV3, and EfficientNetV2. Our method ARRN remains more robust at lower resolution than these standard methods while also beating their performance at the highest resolution. We also include the inference time of all methods across different resolutions for transparency. While our implementation could be improved upon, ARRN achieves a reasonable inference time relative to prior methods that have had years of tuning from the community.
>
> The revised version of our work is also edited to be more concise, and includes a more complete related works section that helps situate and differentiate our contribution relative to prior works.
>
> We hope our comment answers your questions, and that you find the added experiments improve the quality of our work.

---

> > ### Author Response · Authors · 2023-11-20
> >
> > Dear Reviewer,
> >
> > We hope that you've had a chance to read our responses and clarification. As the end of the discussion period is approaching, we would greatly appreciate it if you could confirm that our updates have addressed your concerns.

---

> > > ### Author Response · Authors · 2023-11-23
> > >
> > > Dear reviewer,
> > >
> > > We have just provided a second revision of our experiments that includes the higher-resolution STL-10 dataset. Our method ARRN retains a significant advantage in accuracy at all resolutions against all baselines.
> > >
> > > As the discussion period is coming to an end, we would appreciate that you clarify whether the revised manuscript, improved experiments, and answers to your questions have addressed your concerns.

---

### Meta-Review · Area_Chair_82Jy · 2023-12-06

**Metareview:**

This paper propose adaptive resolution residual network that can apply a single network at various resolution. The main technique is based on laplacian pyramids and dropping out the laplacian residuals during training time as augmentation. The paper is well written and the approach seems intuitive and supported by earlier works in signal processing.

The main concern of the paper is the experiments. Reviewers would like to see direct comparisons with other methods that can handle adaptive resolution. Additionally, data augmentation of different resolution via resizing should be applied to baselines as well. Next, the choice of the tasks and dataset is not suitable for evaluation the proposed method. Specifically, the choice of CIFAR dataset and the STL in the rebuttal are very low in resolution to beging with. Lastly, tasks beyond classification are probably more suitable to demonstrate the effective of the approach. For example, segmentation would be more suitable where the output also dependent on the input resolution.

**Justification For Why Not Higher Score:**

Reviewers raise concerns on the experiment and compared baselines.

**Justification For Why Not Lower Score:**

N/A

---

### Decision · Program_Chairs · 2024-01-16

Reject